# Postmortem Stability Analysis of Lipids and Polar Metabolites in Human, Rat, and Mouse Brains

**DOI:** 10.3390/biom15091288

**Published:** 2025-09-05

**Authors:** Marina Zavolskova, Dmitry Senko, Olga Bukato, Sergey Troshin, Elena Stekolshchikova, Mark Kachanovski, Anna Akulova, Maria Afonina, Olga Efimova, Daria Petrova, Maria Osetrova, Philipp Khaitovich

**Affiliations:** 1Skolkovo Institute of Brain-Inspired Technologies, Bolshoy Boulevard, 30, Bld. 1, Moscow 121205, Russia; 2Institute of Clinical Morphology and Digital Pathology, I.M. Sechenov First Moscow State Medical University, Abrikosovsky Lane, 1, Moscow 119435, Russia

**Keywords:** mouse, rat, human, brain, lipidome, metabolome, stability, post-mortem interval, UPLC-MS/MS

## Abstract

Lipids and polar metabolites are emerging as promising indicators of the brain’s molecular phenotype in both clinical and fundamental research. However, the impact of postmortem delay on these compounds, unavoidable in human brain studies, remains insufficiently understood. In this study, we examined the postmortem stability of lipids and polar metabolites over a 48-h interval in the brains of three species: humans, rats, and mice. We show that the abundance levels of 23% of the 867 studied lipids and 75% of the 104 studied polar metabolites were affected significantly by postmortem delay in at least one species. The postmortem effects correlated positively and significantly among the species, while showing an approximately tenfold slower rate in humans compared to rodents. The only exception to the postmortem rates deceleration was a group of oxidized fatty acids, which accumulated at similar speed in both humans and rodents. These findings provide valuable insights for improving reproducibility and refining the interpretation of human and rodent brain lipidome and metabolome data in future studies.

## 1. Introduction

The number of studies investigating the roles of low molecular weight compounds, lipids, and polar metabolites in the human brain is rapidly increasing. This growth is driven partly by advancements in instrumentation and analytical methodologies and partly by a growing recognition of the critical biological roles these compounds play in brain function. Across tissues, these molecules enable energy storage and transfer [1,2], act as primary structural components of cellular membranes [3], regulate membrane flexibility and permeability [4], and serve as signaling molecules [5,6]. In the brain, these functions become more specialized, reflecting its complex organization. For example, the lipid composition of membranes varies significantly between compartments, enabling and modulating brain functionality. Among such compartments are specialized myelin membranes, which form a protective sheath around nerve fibers to accelerate electrical impulse transmission [7], the dynamic membranes of neuronal synapses [8], and the distal and proximal parts of dendritic projections of neurons [9,10]. The lipid composition of these structures has been shown to influence membrane geometry and fluidity, characteristics essential for the functionality of key membrane-bound processes like synaptic transmission and synaptic potentiation [11,12]. Beyond their structural roles in supporting the brain’s functional architecture, lipids also contribute directly to brain signaling. They act as retrograde neurotransmitters (e.g., endocannabinoids) [13], secondary messengers (e.g., diacylglycerides) [14], and mediators of neuroinflammation, such as prostaglandins and leukotrienes [15].

The role of polar metabolites, small, low molecular weight polar compounds, in the brain is even better established. These include small-molecule neurotransmitters and neuromodulators (e.g., glutamate, serotonin, and dopamine) [16], energy molecules (e.g., glutamate, glucose, and ATP) [17,18], secondary messengers (e.g., cAMP and IP_3_) [19], and diverse metabolic intermediates [20], among others. Maintaining physiological levels of lipids and polar metabolites in brain tissue is crucial, as evidenced by human studies showing their substantial alterations in psychiatric and cognitive disorders [21,22]. Furthermore, genetic studies have linked mutations affecting the enzymes involved in lipid and polar metabolite biosynthesis and degradation to these disorders [23,24]. Animal model experiments also highlight the critical role of specific metabolic enzymes in behavioral regulation and susceptibility to disorders [25].

While animal experiments can be tightly controlled, studies of the human brain often rely on opportunistic collection of postmortem samples with varying postmortem delay intervals. A substantial body of research has been conducted to evaluate the effects of postmortem delay and agonal state on the preservation of total RNA and messenger RNA (mRNA), given the intrinsic instability of this biopolymer in the neutral or acidic environment of postmortem tissue and its widespread use in transcriptome analyses [26]. In addition to RNA degradation dynamics, postmortem studies of brain tissue have revealed changes in cell size [27], grey and white matter density [28], and proteome stability [29]. Notably, some of these changes happen at the very early stages of postmortem interval. For instance, mRNA deterioration and neurochemistry alterations have been observed in rats within the first half hour after death [30].

In contrast to the extensive work on RNA and, partially, on protein postmortem stability, studies investigating the effects of postmortem delay on brain lipids and polar metabolites remain limited. In humans, one study examined the stability of 25 polar metabolites, 18 of which were amino acids, over a 4-h interval, identifying eight stable compounds [31]. A parallel analysis in rat brains over a 48-h postmortem interval yielded similar results, narrowing the list of stable metabolites to five. Another study investigated the stability of 63 polar metabolites in the rat brain over a 72-h postmortem period, revealing that 31 compounds exhibited significant alterations after death [32]. Two studies explored postmortem lipid stability, both focusing on the rat brain samples and, therefore, exploring rather short postmortem delay intervals relevant to animal model experiments. The first study analyzed the stability of two lipid classes, diacylglycerols and free fatty acids, demonstrating an accumulation of these lipids within a 30 min postmortem interval [33]. The second study analyzed a broader spectrum of compounds, both polar and nonpolar, in the rat hippocampus up to one hour postmortem [34]. This analysis identified 32 compounds showing significant postmortem alterations, with most showing accumulation over time.

These findings clearly demonstrate that the abundance levels of certain lipids and polar metabolites are influenced by postmortem interval (PMI) duration. However, existing research has notable limitations, particularly regarding human postmortem samples and the range of compounds analyzed over extended postmortem intervals typically associated with human brain sample collection. To address these knowledge gaps, we profiled the stability of lipids and polar metabolites in neocortical samples from three species, humans, rats, and mice, in the parallel time-series experiments. By utilizing surgery-derived samples, we were able to analyze compound levels starting from zero, and up to 48 h postmortem, across all three species. Furthermore, employing untargeted mass spectrometry coupled with high-performance liquid chromatography allowed us to assess a broad spectrum of polar and nonpolar metabolites (lipids), encompassing more than 58 biochemical classes in total.

## 2. Materials and Methods

### 2.1. Postmortem Brain Samples

All animals were group housed in standard conditions of 21–23 °C and 30–40% relative humidity with a 12-h light:12-h dark cycle (light on 8:00 AM) and provided standard chow and water ad libitum in the animal house of the Institute of Higher Nervous Activity and Neurophysiology of the Russian Academy of Science. All experimental procedures followed Directive 2010/63/EU of the European Parliament and of the Council of 22 September 2010 on the protection of animals used for scientific purposes, and were approved by the Institute of Higher Nervous Activity and Neurophysiology of the Russian Academy of Science.

C57BL/6 male mice aged 6 weeks (*n* = 6) were sacrificed by cervical dislocation and decapitated. Wistar male rats aged 7–8 months (*n* = 6) were anesthetized with intraperitoneal injection of 0.5 mg/kg Zoletil 100 (tiletamine and zolazepam mixture) with 0.05 mL/kg Xyla 2% (xylazine); after 10 minutes, eye blink and pedal reflexes were checked, and the rats were decapitated using a guillotine. For both rodent species, brains were immediately removed from the skull, placed on an ice-cold glass plate, and the first frontal lobe neocortical sample dissected and frozen on dry ice (zero time point). The remaining frontal lobe neocortex samples were dissected into approximately 10 mg pieces, which were placed into ice-cold 1.5 mL microtubes and stored at +4 °C until frozen on dry ice at the defined time points of 2, 4, 8, 12, 24, and 48 h after dissection. The samples were then stored at −80 °C until metabolite extraction.

The human brain samples were sourced from patients undergoing standard neurosurgical procedures performed for primary and secondary brain tumors, including metastatic brain lesions and low-grade gliomas. All tumors were completely resected to healthy tissue margins, with approximately 200 mg of visually typical neocortical tissue located at tumors periphery collected for the study. The included donor’s cohort consisted of six patients, four women and two men, with a mean age of 56 years (50–65 years range), with more complete clinical and demographic data summarized in Appendix A. Within 10 min of neocortical tissue excision, each donor sample was divided into 10–20 mg pieces, which were placed into ice-cold 1.5 mL vials and stored at 4 °C until frozen on dry ice at the defined time points of 2, 4, 8, 12, 24, and 48 h after dissection. The samples were then stored at −80 °C until metabolite extraction. All procedures adhered to the Declaration of Helsinki and relevant national ethical guidelines. Informed consent for participation in the study was obtained from each patient. Patient confidentiality was maintained, and tissue collection was performed as part of clinically indicated surgical procedures to ensured minimal risk to participants.

### 2.2. Lipid and Polar Metabolite Extraction

The extraction buffer (MeOH (Carlo reagents, CAS 67-56-1):MTBE (Sharlab, CAS 1634-04-4), (1:3 (*v*/*v*)) was enriched with internal standards including 0.5 µg/mL triacylglycerol (TAG 15:0/18:1-d7/15:0, Avanti Polar Lipids, 791648C), diacylglycerol (DAG 15:0/18:1-d7, Avanti Polar Lipids, 791647C), ceramide (Cer d18:1-d7/15:0, Avanti Polar Lipids, 860681), lysophosphatidylcholine (LPC 18:1-d7, Avanti Polar Lipids, 791643C), phosphatidylglycerol (PG 15:0/18:1-d7, Avanti Polar Lipids, 791640C), phosphatidylcholine (PC 15:0/18:1-d7, Avanti Polar Lipids, 791637C), phosphatidylethanolamine (PE 15:0/18:1-d7, Avanti Lipids, 791638C), methionine-methyl-13C, d3 (Aldrich 299154), and D-L-Aspartic-acid-4 C13 (Aldrich 488968). The buffer was prepared and cooled to −20 °C, serving as a universal extraction agent for a series of samples. The samples were randomized and organized so that the mass spectrometry batch consisted of 48 samples. To control extraction procedure extraction quality controls (EQC) were made from pooled homogenized brain samples and placed in the very beginning, after each 12th sample, and after the last samples. To ensure accuracy, eight blank samples were included, which consisted of tubes without any tissue specimens. Each tube, including the blanks, received 0.5 mL of the pre-cooled extraction buffer. The tissue pieces were then homogenized using a Precellys Evolution Tissue Homogenizer, and then an additional 0.5 mL of extraction buffer was added. After vortexing, the homogenates underwent a 30-min incubation at 4 °C on an orbital shaker, followed by a 10-min ultrasound treatment in an ice-cooled sonication bath. Subsequently, the homogenate was transferred to 2 mL Eppendorf tubes, and a 700 μL mixture of water and methanol (3:1, *v*/*v*) was added. Following a 5-min incubation at 4 °C on an orbital shaker, the homogenates were centrifuged at 12,700 g for 10 min to separate the hydrophobic and aqueous phases. The upper phase, rich in hydrophobic compounds, primarily lipids, was carefully isolated by transferring 540 μL to a 1.5 mL Eppendorf tube. The solvent was then evaporated using a Speed Vac concentrator at room temperature. For quality control (QC) purposes, 5 µL of the upper phase from each sample was collected and pooled. Finally, the lipid and polar extracts were stored at −80 °C until the subsequent mass spectrometry (MS) analysis stage.

### 2.3. UPLC-MS Analysis

The experiments were conducted using a QExactive mass spectrometer equipped with a heated electrospray ionization source from Thermo Fisher Scientific, Waltham, MA, USA, coupled with a Waters Acquity UPLC (version 1.65) chromatographic system from Waters, Manchester, UK. Liquid chromatography (LC) separation was achieved using an ACQUITY UPLC BEH C8 reverse phase column (2.1 × 100 mm, 1.7 μm, Waters Co., Milford, MA, USA) with a Vanguard guard pre-column of the same material kept at 60 °C for non-polar phase and with an ACQUITY UPLC BEH HILIC column (2.1 × 150 mm, 1.7 μm, Waters Co., Milford, MA, USA) with a Vanguard guard pre-column of the same material kept at 45 °C for polar phase.

### 2.4. Lipids

To resuspend the dried lipid pellets, each sample was treated with 200 µL of ice-cold acetonitrile (Honeywell, Charlotte, NC, USA CAS 75-05-8):isopropanol (Fisher Chemical, Waltham, MA, USA, CAS 67-63-0) solution (7:3, *v*:*v*). The samples were vortexed, shaken (10 min), sonicated (ice-cooled, 10 min), and centrifuged (5 min at 12,700 g). The supernatant was then transferred to a new 1.7 mL Eppendorf tube. Before mass spectrometry analysis, 25 µL of the lipid sample was transferred to a 2 mL autosampler vial and diluted 1:5 or with acetonitrile:isopropanol solution (3:1 (*v*:*v*) for positive and negative ion measurements, respectively. A pooled sample was prepared for quality control (QC) and injected multiple times throughout the analysis: five times at the beginning of the batch to condition the column and then after each 12th sample of the batch.

Mobile phase A for the chromatography contained 10 mM ammonium acetate (Sigma-Aldrich, St. Louis, MO, USA, CAS 64-19-7) in water (SCHARLAB, Barcelona, Spain, CAS 7732-18-5) with 0.1% formic acid (Merck, CAS 84-18-6), while mobile phase B consisted of 10 mM ammonium acetate in a mixture of ACN and IPS in a ratio of 7:3 with 0.1% formic acid. In the negative detection mode, acetic acid (Fisher Chemical, CAS 64-19-7) was used instead of formic acid in the composition of the mobile phase. The chromatography flow rate was 0.4 mL/min, and a gradient elution mode was used. The program included specific time intervals and linear changes in mobile phase B composition. The program consisted of: 1 min at 55% B, a 3-min linear gradient from 55% to 80% B, an 8-min linear gradient from 80% to 85% B, and a 3-min linear gradient from 85% to 100% B. The phase composition was maintained at 100% B for 4.5 min, followed by a return to the initial conditions (55% B) for 4.5 min. The injection volume was 3 μL. MS detection was conducted to register both positively and negatively charged ions within the *m*/*z* range of 100 to 1500, with a mass resolution of 70,000 (FWHM for *m*/*z* 200). The ESI source conditions were set as follows: the flow rate of the dryer gas was 4 c.u., the flow rate of the auxiliary gas was 20 c.u., and the gas flow rate at the atomizer was 45 c.u. The voltage on the capillary was set to 4.5 kV in positive mode and −3.5 kV in negative mode. The nebulizer temperature was maintained at 350 °C, while the capillary temperature was set to 250 °C. The S-lens RF level parameter had a value of 70. The maximum ion accumulation time in the trap was 100 ms, and the maximum amount of accumulated ions was 1 × 10^6^. Before sample analysis, external calibration was performed using the Pierce LTQ Velos ESI positive ion calibration solution and Pierce ESI negative ion calibration solution to ensure mass accuracy.

Lipid fragmentation was performed using an iterated data-dependent acquisition (DDA) approach. The protocol involved conducting an analytical MS/MS analysis with inclusion lists that contained around 400 *m*/*z* values for each polarity. This process was then repeated with an exclusion list containing the same *m*/*z* values. Parameters for the full MS scan mode were set as follows: a resolution of 70,000 at *m*/*z* 200, an AGC target of 5 × 10^5^, an IT of 50 ms, and a mass range of 200–1800. In the fragmentation step, ions from the inclusion list within a 10 ppm range were selected. Fragmentation was carried out with a resolution of 17,500, an AGC target of 5 × 10^4^, an IT of 100 ms, an intensity threshold of 8 × 10^3^, and an isolation width of 1.2 Da. Stepped normalized collision energy was set at 15%, 20%, and 25%. A dynamic exclusion parameter of 12 s was applied. Data was acquired in profile mode, and isotope exclusion was enabled.

### 2.5. Polar Metabolites

The preparation of polar metabolite samples was similar to that of lipid samples, except for the resuspend solution (ice-cold acetonitrile (Honeywell, CAS 75-05-8):H_2_O (4:1, *v*:*v*)) and volume of sample in the vial (30 µL). Quality control samples were prepared identically.

Mobile phase A for polar phase analysis contained 10 mM ammonium formate (sigma-aldrich, CAS 64-19-7) in a mixture of H_2_O and ACN in a ratio of 95:5 with 0.1% formic acid, while mobile phase B consisted of 10 mM ammonium formate (sigma-aldrich, CAS 64-19-7) in a mixture of ACN and H_2_O in a ratio of 95:5 with 0.1% formic acid. The chromatography flow rate was 0.4 mL/min, and a gradient elution mode was used. The program included specific time intervals and linear changes in mobile phase B composition. The program consisted of: 0.5 min at 100% B, a 7.2-min linear gradient from 100% to 70% B, a 1.8-min linear gradient from 70% to 40% B, a 0.75-min linear gradient from 40% to 30% B, and a 1-min linear gradient from 30% to 0% B. The phase composition was maintained at 0% B for 0.75 min, followed by a return to the initial conditions (100% B) for 1 min with a flow rate of 0.6 mL/min. The injection volume was 6 μL. MS detection was conducted to register both in polarity switch regime within the *m*/*z* range of 100 to 650, with a mass resolution of 70,000 (FWHM for *m*/*z* 100). The ESI source conditions were set as follows: the flow rate of the dryer gas was 4 c.u., the flow rate of the auxiliary gas was 20 c.u., and the gas flow rate at the atomizer was 45 c.u. The voltage on the capillary was set to 4.5 kV in positive mode and −3.5 kV in negative mode. The nebulizer temperature was maintained at 350 °C, while the capillary temperature was set to 250 °C. The S-lens RF level parameter had a value of 70. The maximum ion accumulation time in the trap was 100 ms, and the maximum amount of accumulated ions was 1 × 10^6^. Before sample analysis, external calibration was performed using the Pierce LTQ Velos ESI positive ion calibration solution and Pierce ESI negative ion calibration solution to ensure mass accuracy.

### 2.6. Compound Fragmentation

Lipid fragmentation and polar metabolite fragmentation were both performed using an iterated data-dependent acquisition (DDA) approach. For lipid fragmentation, the analytical MS/MS analysis involved inclusion lists containing approximately 400 *m*/*z* values for each polarity, which were subsequently repeated with exclusion lists of the same values. The parameters for the full MS scan mode included a resolution of 70,000 at *m*/*z* 200, an AGC target of 5 × 10^5^, an IT of 50 ms, and a mass range of 200–1800.

In the fragmentation step for lipids, ions from the inclusion list within a 10 ppm range were selected. Fragmentation was conducted at a resolution of 17,500, with an AGC target of 5 × 10^4^, an IT of 100 ms, an intensity threshold of 8 × 10^3^, and an isolation width of 1.2 Da. The stepped normalized collision energy was set at 15%, 20%, and 25%, with a dynamic exclusion parameter of 12 s. Data acquisition was performed in profile mode, with isotope exclusion enabled.

Similarly, polar metabolite fragmentation also utilized an iterated DDA approach, but with inclusion lists containing around 200 *m*/*z* values for each polarity, followed by exclusion lists with the same values. The full MS scan parameters for polar metabolites were set with a resolution of 70,000 at *m*/*z* 100, an AGC target of 1 × 10^6^, an IT of 100 ms, and a mass range of 85–650.

During the fragmentation step for polar metabolites, ions from the inclusion list within a 10 ppm range were selected, with fragmentation carried out at a resolution of 17,500, an AGC target of 5 × 10^5^, an IT of 50 ms, an intensity threshold of 8 × 10^3^, and an isolation width of 1.2 Da. The stepped normalized collision energy was set at 20%, 30%, and 40%, with a dynamic exclusion parameter of 6 s. Data was acquired in profile mode, and isotope exclusion was enabled.

### 2.7. MS Data Extraction and Preprocessing

After acquiring UPLsC-MS spectra, the data were processed with MS-DIAL software (version 4.90) in separate positive and negative mode projects. Specific parameters were set, including MS1 tolerance (0.05), minimum peak height (10,000), and mass slice width (0.05). A lipid annotation was performed using MS-Dial database, an internal lipid database containing internal standards information, and confirmed manually to check integration quality. It included 37 lipid classes (PC, PE, PC-O, PE-O, PG, FA, DG, CAR, LPC, Cer, CL, TG, PE-P, HexCer, SM, SHexCer, LPE, PI, MGDG, NAE, PS, OxFA, LPE-O, BMP, CoQ, AHexCer, LPS, LPI, TG-O, PMeOH, HBMP, ST, PS-O, LPC-O, GM, Cholesterol, and CE) and 21 polar metabolite classes. The results were exported as a “Raw data area” matrix and converted to .csv format for further R processing.

To merge the polarities and filter duplicate lipids, retention times were correlated. The positive polarity was given preference during the filtration process. Zero values were replaced with 0.25 times the corresponding lipid’s minimal area, and each value was multiplied by a random number between 0.9 and 1.1. A log_2_-transformation was then applied to all values.

To exclude chemical contamination and technical noise, filtering procedures were implemented. First, a filter based on blank samples was applied, selecting only features with mean intensity at least four times higher in samples compared to blanks. A variance filter was then used, selecting only peaks with a standard deviation below 0.5 across QC samples. Lipid intensities were further normalized based on the median value of standards in a sample and the wet weight of the sample. Wet weights and standard intensities were log_2_-transformed, and sample values were subtracted from the mean of the corresponding parameter.

### 2.8. Statistical Analysis

Metabolite intensity data were normalized to the donor mean across all measured lipids and polar metabolites to account for inter-individual variability. To visualize the global effect of freezing time on metabolite profiles, PCA was performed separately for each species (mice, rats, humans) using the prcomp() function from the R stats package. To evaluate metabolite stability over time, we performed correlation analysis for each metabolite: for each metabolite, a linear model was fitted between normalized intensity values (from 6 donors) and the log_2_-transformed freezing time (0, 2, 4, 8, 12, 24, and 48 h). Resulting *p*-values were adjusted using the Bonferroni method to control the family-wise error rate. Metabolites with an adjusted *p*-value < 0.05 were classified as unstable. To assess cross-species consistency in degradation trends, pairwise Pearson correlations were computed between the regression coefficients (slopes) of metabolites across species. Unstable metabolites were further analyzed to identify common degradation patterns. Normalized intensity trends over time were clustered using the k-means algorithm (k = 3 for lipids, k = 2 for polar metabolites). The optimal number of clusters was determined via the elbow method. Clusters were labeled based on their degradation dynamics (e.g., “Decreasing” and “Increasing” for directed changes). To determine whether specific lipid classes or structural features were associated with stability, the enrichment of lipid classes and unsaturation level in each cluster was assessed using hypergeometric distribution, followed by Benjamini-Hochberg (BH) correction for multiple comparisons. In a study where time intervals coincided with those discussed in this study, the intensity of correlations between this work and the existing literature was assessed using the Spearman rank correlation method. To evaluate the randomness of the observed positive shift in correlations, a permutation test was conducted, involving 10,000 random shuffles. In a study where direct correlation of changes over time was not feasible, a Fisher’s exact test was conducted to assess the direction of change between the two studies. All statistical analyses were conducted using R language and a significance level of *p* < 0.05 was considered statistically significant for all tests unless otherwise noted.

## 3. Results

To evaluate the effects of postmortem delay on the abundance levels of polar and hydrophobic metabolites (lipids), we collected brain tissue samples from human brain surgery cases (*n* = 6), as well as from laboratory rats and mice (*n* = 6 each). All samples were taken from the temporal lobe of the cerebral cortex (Appendix A). The sample aliquots were then snap frozen either immediately after resection (zero time point) or at 2, 4, 8, 12, 24, and 48 h post-surgery or post-sacrifice (Figure 1A).

We extracted lipids and polar metabolites from the 126 frozen brain aliquots and analyzed their abundance levels using high performance liquid chromatography coupled with tandem mass spectrometry (UPLC-MS), with separate measurements conducted for lipids and polar metabolites in both positive and negative ionization modes. Following the mass spectrometric analysis, we identified the intensities of 874 lipids and 104 polar metabolites, annotated based on their chromatographic mobility, molecular ions, and fragmentation masses (Figure 1B,C; Appendix A). The lipids encompassed 37 distinct lipid classes, representing most major brain lipid components, while the polar metabolites were categorized into 21 functional classes based on their chemical groups (Figure 1B,C).

General variation assessment, conducted using principal component analysis (PCA) based on the abundance levels of all 874 detected lipids (Figure 2A) or all 104 detected polar metabolites (Figure 2B), revealed a distinct separation of samples by collection time point along the first principal component (PC1) in mice and rats, indicating a substantial influence of postmortem delay duration in these species. In humans, however, separation according to postmortem time was far less pronounced for both polar and non-polar compounds. Consistent with the PCA results, statistical analysis yielded substantially more compounds significantly affected by postmortem interval in mouse and rat brains compared to humans. Specifically, linear regression analysis of normalized compound intensities and postmortem interval duration identified 166, 135, and five lipids and 58, 58, and five polar metabolites significantly affected by postmortem interval in mice, rats, and humans, respectively (*p* < 0.05, Bonferroni corrected; Appendix A; Appendix A). Combined, 23% of detected lipids and up to 75% of detected polar metabolites exhibited significant abundance level dependence on postmortem interval duration in at least one species (postmortal interval unstable lipids and polar metabolites; *p* < 0.05, Bonferroni corrected; Appendix A).

Despite the apparent distinction between rodents and humans in postmortem effect amplitude, the postmortem changes in lipid and polar metabolite abundance levels showed significant positive correlations between all species pairs, including rodent-human comparisons (Pearson correlation, r (lipids: rats ~ mice) = 0.76, *p* < 0.0001, r (lipids: humans ~ rats) = 0.12, *p* < 0.0001, r (lipids: humans ~ mice) = 0.23, *p* < 0.0001; r (polar metabolites: rats ~ mice) = 0.78, *p* < 0.0001, r (polar metabolites: humans ~ rats) = 0.4, *p* < 0.0006, r (polar metabolites: humans ~ mice) = 0.42, *p* < 0.0001; Figure 2C,D). This finding suggests that postmortem level alterations are shared among humans, rats, and mice, which aligns with our knowledge of the shared biochemical properties of polar and hydrophobic metabolites in these species. The smaller amplitude of postmortem interval effects observed in the human brain could potentially be explained by previously reported differences in postmortem tissue deterioration rates between human and rodents—a point discussed in detail later in text.

To investigate whether the postmortem delay affects particular biochemical classes of lipids and polar metabolites in a specific manner, we performed unsupervised clustering of their delay-dependent profiles. The analysis identified three distinct clusters of delay-dependent trajectories for lipids and two for polar metabolites, in addition to a cluster containing compounds with stable, delay-independent profiles (Figure 3; Appendix A). For both lipids and polar metabolites, two primary trends were, unsurprisingly, a monotonic increase and a monotonic decrease in abundance levels along the postmortem interval (increasing clusters (IC) and decreasing clusters (DC); Figure 3A,B). For lipids, however, we identified a unique cluster comprising six compounds that exhibited a much more rapid increase in abundance levels with prolonged postmortem delay in all three species (second increasing cluster or IC2).

A formal comparison of postmortem delay-induced changes of lipid and polar metabolite abundance levels, represented as the slopes of linear regression trajectories fitted to the data, revealed several differences in alteration rates both between lipids and polar metabolites and across species. On average, the three species exhibited approximately 2.5-fold faster degradation rates for polar metabolites compared to lipids (metabolic DC and lipid DC; Figure 3C). The accumulation rates of most lipids were comparable to those of polar metabolites, with the notable exception of six rapidly accumulating lipid compounds (metabolic IC and lipid IC1 and IC2; Figure 3C). These patterns were also evident at the individual species level (Appendix A). Furthermore, consistent with the clustering and statistical analysis results, human lipids and polar metabolites displayed significantly slower postmortem degradation and accumulation rates compared to rodents, with the sole exception of six rapidly accumulating lipid compounds (lipid IC2). For these compounds, the accumulation rates in humans and mice were similar, while rats exhibited somewhat slower accumulation pace, albeit still substantially faster than lipids comprising IC1 cluster (Appendix A).

Analysis of the biochemical properties of stable, semi-stable (DC and IC1), and unstable (IC2) lipids revealed significant differences among these groups. Specifically, all IC2 lipids belonged to a single biochemical class: oxidized fatty acids (OxFA; hypergeometric test, BH-corrected *p* < 0.05). In contrast to unstable lipids, semi-stable ones were enriched in free fatty acids (FA), lysophosphatidylcholine (LPC), triglycerides (TG), and phosphatidylglycerol (PG). Stable lipids, on the other hand, were enriched in the most abundant membrane lipid classes: phosphatidylcholines (PC) and phosphatidylethanolamines (PE) (hypergeometric test, BH-corrected *p* < 0.05; Figure 4A). In addition to lipid class distribution, we examined the relationship between postmortem delay stability and general structural properties of lipid compounds, such as the length and unsaturation of their fatty acid residues. While fatty acid tail length did not significantly differ among the three stability groups, the degree of unsaturation was important. Saturated fatty acids were more prevalent in stable lipids, oligo-unsaturated fatty acids were predominant in semi-stable lipids, and poly-unsaturated fatty acids tended to be more present in unstable lipid group (Figure 4B).

In contrast to lipids, polar metabolites did not exhibit any significant differences in biochemical or metabolic pathway properties across the three clusters. To validate the observed postmortem delay-dependent changes in polar metabolite abundance levels, we took advantage of data from studies examining the stability of polar metabolites in postmortem brain tissue from humans, rats, and mice [31,32]. Of the 28 metabolites analyzed in study [31], 20 overlapped with metabolites from our dataset. For 17 out of these 20 metabolites, delay-dependent changes showed positive correlations between the two datasets, significantly exceeding the chance expectation (binominal test, *p* = 0.002; permutation test, *p* = 0.013). In study [32], among 31 metabolites that significantly changed over 72 h postmortem, 16 overlapped with those identified in our study. Of these, 11 showed the same direction of postmortem change, significantly more than expected by chance (Fisher test, *p* = 0.035; Appendix A).

## 4. Discussion

This study examines the stability of lipids and polar metabolites in the brains of three species, humans, rats, and mice, in relation to the postmortem interval. Expanding the scope of prior research, this work provides a more comprehensive assessment of lipid and polar metabolite postmortem stability in brains of humans, rats, and mice over a 48-h period. For all three species, we monitored postmortem changes in the abundance levels of 874 lipids and 104 polar metabolic compounds. The key findings of this study can be summarized as follows: First, we show that the abundance levels of 673 lipids and 24 polar metabolites remain stable in postmortem brain samples for up to 48 h. Thus, brain studies focusing on these compounds are unlikely to be affected by postmortem artifacts. Second, for the remaining and 201 assessed lipids 80 polar metabolites, we documented patterns of abundance changes over the 48-h postmortem period both in humans and in model rodent species. This dataset provides detailed information for more accurate interpretation of metabolic analyses in postmortem brain studies. As outlined in the results section, two key general trends of the observed postmortem alteration patterns include: (i) a high degree of consistency of these profiles across the three species, and (ii) significantly slower rates of alteration in humans compared to rodents for all unstable compounds, with the notable exception of oxidized free fatty acids. In the discussion below, we delve into these interspecies differences as well as the biochemical and functional characteristics of both stable and unstable compounds that may underly the observed postmortem stability patterns.

Based on existing knowledge, several factors might contribute to the difference in polar metabolite and lipid stability rates between human and rodent postmortem brain samples. Humans have a slower metabolic rate than rats, resulting in a more gradual depletion of brain energy reserves postmortem and a delayed onset of decomposition. In contrast to humans, rats and mice, with their higher metabolic rates, experience faster energy depletion and subsequent cellular changes [35,36]. Supporting this notion, previous postmortem stability studies of enzymes, messenger and microRNAs, and histone modifications demonstrated a greater stability of these biomolecules in human brain tissue compared to that of rats [34,37,38,39]. Finally, the larger size of the human brain might contribute to slower metabolite diffusion and decomposition rates [32,35].

Among lipids, however, one type of compounds, oxidized fatty acids, showed the same rapid accumulation rates in humans and mice, with slower but still rapid accumulation rates in rats. Oxidized fatty acids are free fatty acids containing one or more hydroxy groups. Formation of these compounds can be driven by reactive oxygen species (ROS) through formation of lipid hydroperoxides, which can decompose into aldehydes and other reactive carbonyl compounds. After death, the absence of antioxidant defenses leads to the rapid oxidation of polyunsaturated fatty acids, resulting in various oxidized lipid species, a process further accelerated in hypoxic or ischemic conditions [40,41,42]). Our results indicate that conditions of human and mouse brains support this type of oxidation-driven postmortem artifacts to the same extent. Notably, while the cluster comprised of rapidly accumulating lipids (lipid IC2) consists only of oxidized fatty acids, the slowly increasing cluster (lipid IC1) also contains one oxidized fatty acid. Comparison of biochemical properties of this fatty acid to the six rapidly accumulating lipids showed that, unlike in the six, its hydroxy group localized next to the fatty acid tail bend, potentially deselecting oxidation rate (Appendix A). More generally, our results show that the structural properties of lipids’ fatty acid residues protect them from oxidation, contributing to the greater postmortem stability of fully saturated residues, while lipids containing polyunsaturated fatty acid residues tend to be unstable, aligning with their greater susceptibility to oxidation [43]. Most non-oxidized free fatty acids also show an accumulation effect in our study, most likely due to phospholipid hydrolysis. In particular, stearic acid and palmitic acid have already been proposed as indicators of the postmortem interval [44]. However, our results show that levels of abundant oxidized fatty acids recorded in our study might be a more effective indicator of the postmortem delay duration.

In addition to oxidized and free fatty acids, our study revealed particular postmortem stability patterns for other biochemical classes of lipids. Phosphatidylcholines (PC) and phosphatidylethanolamines (PE), the main structural components of cell membranes, showed the greatest stability over 48 h postmortem. This is a positive result, suggesting that membranes of brain cells, one of the key elements of signal transduction and cell communication processes, retains its main properties in the postmortem tissue. Lipids comprising lysophosphatidylcholines (LPC), LPE, triglycerides (TG), and phosphatidylglycerols (PG) were found to be semi-stable. LPE and LPCs are intermediates of phospholipid metabolism and can be formed by the hydrolysis of PC and PE by phospholipases. Their “semi-stability” may reflect the balance between the processes of formation and degradation. On the one hand, the activity of phospholipases can increase postmortem, leading to the formation of LPC from more stable PC and PE. On the other hand, LPCs can also undergo further hydrolysis. Phosphatidylglycerols (PGs) are minor membrane lipids that play a role in mitochondrial membranes but are less common in cellular membranes. While brain studies on changes in this class are lacking, muscle studies show no significant changes in dissected tissue after 24 h, though a degradation trend was observed [45]. Consistently, our data indicates most members of this class show slight postmortem degrade in the brain tissue of the three species, except for PG 42:10, which showed postmortem accumulation.

Postmortem alteration profiles of polar metabolites formed two clusters, one containing degrading and another—accumulating compounds. Among the degrading metabolites, the greatest amplitude of changes were shown by acetylindole, serotonin, and 5′-S-methylthioadenosine. Acetylindole, a component of tryptophan metabolism, is both synthesized and utilized through various pathways within living cells. Its breakdown can occur via hydroxylation at specific sites on the indole ring. Electron-donating groups promote this hydroxylation, whereas electron-withdrawing groups can hinder it, influencing the degradation route [46]. Furthermore, sequential hydroxylation can result in ring cleavage, a crucial step in the breakdown of indole compounds [46]. Postmortem, the degradation of acetylindole is further influenced by increased levels of reactive oxygen species (ROS), which are generated due to mitochondrial dysfunction and oxidative stress occurring after death. ROS can enhance the hydroxylation of acetylindole by promoting oxidative modifications on the indole ring, thereby accelerating its breakdown [47]. Serotonin, a monoamine neurotransmitter and indole derivative, plays a role in tryptophan metabolism. While our data clearly shows degradation of serotonin in the postmortem brain tissue, this finding contrasts some reports. Specifically, study [48] reports a weak but discernible upward trend for serotonin levels in rat brains. This discrepancy might stem from differences in experimental procedures. In a published study, rat heads were left intact post-decapitation, whereas our experiment involved brain extraction. A similar procedure was presented in the study [49], where an increase in serotonin was observed regardless of the inertness of the storage environment and storage temperature. The accumulation effect was evident 4 h after storage. On the other hand, a significant decrease in serotonin levels was detected in a study conducted on rats, rabbits, and humans [50]. The most pronounced reduction in concentration was observed in the hypothalamus, although degradation was also noted in cortical regions. The study further identified the primary cause of these losses as the action of the enzyme monoamine oxidase. Additionally, the researchers examined the rate of degradation in “separated” versus “unseparated” tissues (as defined by the authors), finding that degradation occurred more rapidly in separated tissues—conditions that more closely resemble those in our own work. Overall, postmortem changes in brain serotonin levels vary depending on factors such as time, temperature, and brain region. A key intermediate in the methylthioadenosine (MTA) cycle, also known as the methionine salvage pathway, is 5′-Methylthioadenosine, which is an S-methyl derivative of adenosine. Its decrease may be associated with the destruction of the glycosidic bond [51]. Similarly, other compounds containing a glycosidic bond are degraded, although to a lesser extent. In contrast, the majority of metabolites involved in amino acid metabolism, particularly aspartate, phenylalanine, and others, were found in the ascending cluster. Their accumulation might indicate protein degradation [52,53].

To determine the degree of postmortem change of the compound, we used the Bonferroni correction, which could potentially lead to a type II error. However, using the Benjamini-Hochberg correction in this data defines the compounds as unstable with an absolute change on average 10 times smaller than with the Bonferroni correction. Thus, use of the Bonferroni correction in our statistical analysis allows us to single out compounds, not only showing the statistical significance but also displaying substantial amplitude of postmortem alterations.

The assessment of metabolite and lipid instability in this study is based on samples from six donors per time point. A sample size of six per group appears sufficient to capture key trends in molecular stability, as the applied donor-based normalization procedure helps minimize the influence of inter-individual variability. Consistency of postmortem alterations across all three species, including humans, further supports this approach. As shown in Appendix A, the slopes of degradation and accumulation curves for compounds passing the significance cutoff after multiple testing correction are already modest. Increasing the number of donors beyond six would likely yield statistical significance for compounds with even smaller postmortem shifts, which in many cases may be below the threshold of biological relevance.

An important point for discussion is inter-individual variability. Human samples showed slightly higher variability than rodents, though for most compounds, levels overlapped between species and remained moderate. Donor-based normalization reduced this effect. Importantly, not all compounds demonstrated slower changes in humans; for instance, oxidized fatty acids accumulated rapidly across all three species (Appendix A), indicating that inter-individual variability is not the main factor underlying slower postmortem changes in the human brain.

In this study, rodents were examined alongside humans, as these species are common models in brain research [54]. In rodents, postmortem delay is less problematic due to controlled sample collection, and the time scale studied (up to 48 h) does not reflect typical experimental conditions. Their inclusion primarily enabled comparison of postmortem dynamics in a more controlled model, previously shown to exhibit faster mRNA and protein degradation, with those in humans. Both mice and rats showed faster postmortem dynamics than humans, but the direction of changes was largely consistent across species.

Our study had several limitations. First, the human samples were collected during tumor surgery and represent surrounding brain tissue, which may differ from healthy brain tissue. Nonetheless, the consistency of postmortem alteration traits among the six tested human individuals, as well as the similarities observed between humans and rodents, suggest that the samples used adequately reflect postmortem dynamics of human brain lipids and polar metabolites. Second, although the number of lipids assessed in our study (874) reflects the current state-of-the-art in lipidomics [55,56], certain lipid classes commonly found in the brain, such as eicosanoids, were not included in our analysis, and prenol lipids, gangliosides, cholesteryl esters, and sterol lipids had only one representative each. Similarly, while we quantified more than 100 polar metabolites, our analysis does not comprehensively cover all major metabolic classes [57]. For example, most sugar alcohols, sugar acids, and monosaccharides detected in our study were excluded during the variability screening stage or low annotatie reliability. Third, our study focuses exclusively on postmortem dynamics in the neocortex. There are, however, indications that postmortem degradation rates may vary across brain regions. For instance, the brainstem and thalamus have been reported to exhibit distinct patterns of acetate and inositol changes postmortem [58,59]. One more limitation is that the experimental design does not fully replicate the intact postmortem environment. In postmortem brain studies, the intact human brain typically remains at body, then room temperature for several hours before dissection. Sampling disrupts this state and may alter degradation rates. In this study, storage at +4 °C was chosen as a compromise: it approximates real conditions of non-freezing exposure while limiting excessive degradation caused by dissection. Although this approach cannot fully replicate the intact environment, it provides a controlled framework to assess directional trends and relative stability of lipids and polar metabolites. Another limitation of this study is the absence of direct comparisons of tissue before and after freezing at −80 °C. Nevertheless, snap-freezing and storage at −80 °C are widely recognized as effective for minimizing post-collection biochemical alterations, including oxidation. Evidence from plasma studies indicates high metabolite stability under these conditions, even after long-term storage [60]. In our work, all samples were flash-frozen immediately after dissection and stored for less than one month, further reducing the risk of degradation. Still, systematic evaluation of freezing effects on brain metabolomics is limited and requires further investigation.

## 5. Conclusions

Our study provides a comprehensive characterization of the postmortem instability of brain lipids and polar metabolites in humans and rodents over a 48-h interval. The observed differences in stability across metabolite classes, as well as the influence of fatty acid saturation and structure, underscore the complexity of postmortem changes. These findings lay a foundation for developing strategies to account for the effects of postmortem alterations in brain studies. Specifically, the data generated here could inform future efforts to create correction methods that minimize postmortem alteration effects. Additionally, our results could help refine experimental designs by accounting for the postmortem stability of biochemical classes of lipids and polar metabolites targeted by the study. Ultimately, such steps would enhance the reliability and interpretability of brain metabolomics data obtained from postmortem samples.

## Figures and Tables

**Figure 1 biomolecules-15-01288-f001:**
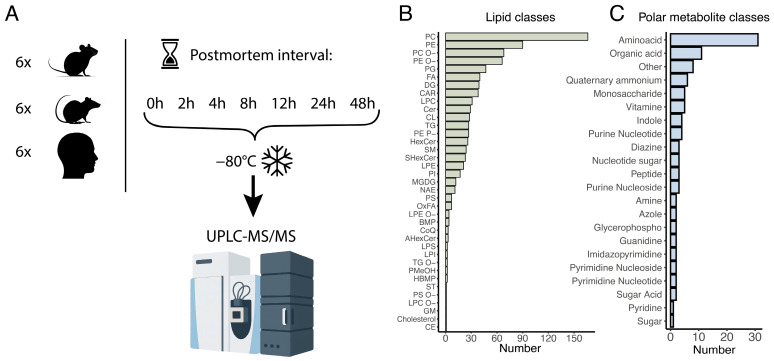
Experiment outline. (**A**). Schematic representation of experimental setup. (**B**). Numbers of detected lipid compounds, grouped according to their biochemical properties. Lipid class abbreviations stand for: AHexCer—acylhexosylceramide; BMP—bis(monoacylglycero)phosphate; CAR—acylcarnitine; CE—cholesteryl ester; Cer—ceramide; CL—cardiolipin; CoQ—coenzyme Q; DG—diacylglycerol; FA—fatty acid; GM—ganglioside; HBMP—hydroxy-bis(monoacylglycero)phosphate; HexCer—hexosylceramide; LPC—lysophosphatidylcholine; LPC-O—lysophosphatidylcholine ether; LPE—lysophosphatidylethanolamine; LPE-O—lysophosphatidylethanolamine ether; LPI—lysophosphatidylinositol; LPS—lysophosphatidylserine; MGDG—monogalactosyldiacylglycerol; NAE—N-acylethanolamine; OxFA—oxidized fatty acid; PC—phosphatidylcholine; PC-O—ether-linked phosphatidylcholine; PE—phosphatidylethanolamine; PE-O—ether-linked phosphatidylethanolamine; PE-P—plasmalogen phosphatidylethanolamine; PG—phosphatidylglycerol; PI—phosphatidylinositol; PMeOH—phosphatidylmethanol; PS—phosphatidylserine; PS-O—ether-linked phosphatidylserine; SHexCer—sulfated hexosylceramide; SM—sphingomyelin; ST—sulfatide; TG—triacylglycerol; TG-O—ether-linked triacylglycerol. A list of the detected lipids is provided in Appendix A. (**C**). Numbers of detected polar metabolite compounds, grouped according to their biochemical properties. A list of the detected polar metabolites is provided in Appendix A.

**Figure 2 biomolecules-15-01288-f002:**
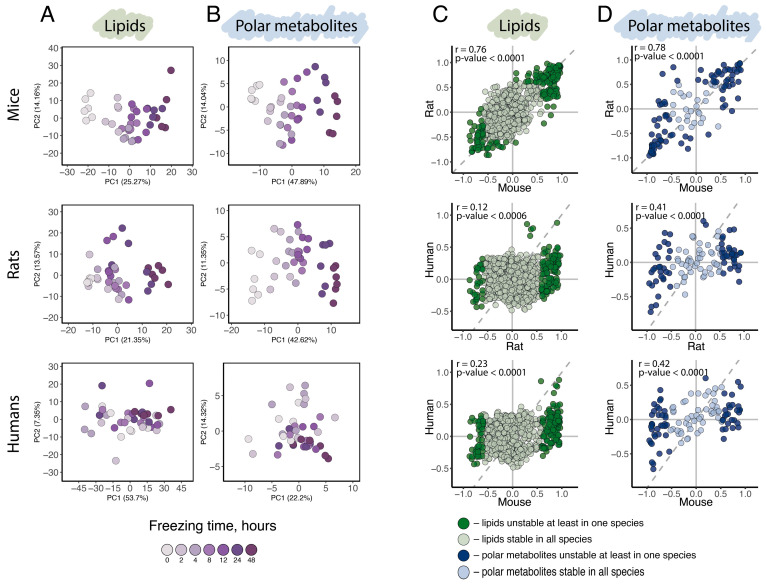
Impact of postmortem interval duration on lipid and polar metabolite abundance levels in mouse, rat, and human brain samples. (**A**,**B**). Visualization of differences in lipidome (**A**) and polar metabolome (**B**) abundance levels among brain samples within each of the three species (top—mice, middle—rats, bottom—humans) using PCA. Each symbol corresponds to a tissue sample. Colors mark freezing time as shown by the legend below the panels. (**C**,**D**). Pairwise comparisons of postmortem changes in lipid (**C**) and polar metabolite (**D**) abundance levels between species. Species names are marked along the axes. The abundance changes for each species are shown as log_2_-transformed differences between zero and 48-h time points in each species. Each symbol corresponds to a compound. Darker shades of color indicate compounds showing statistically significant postmortem alterations in at least one of the species. Pearson correlation coefficients and corresponding *p*-values are plotted above each panel. Dotted line represents the line of equality (y = x).

**Figure 3 biomolecules-15-01288-f003:**
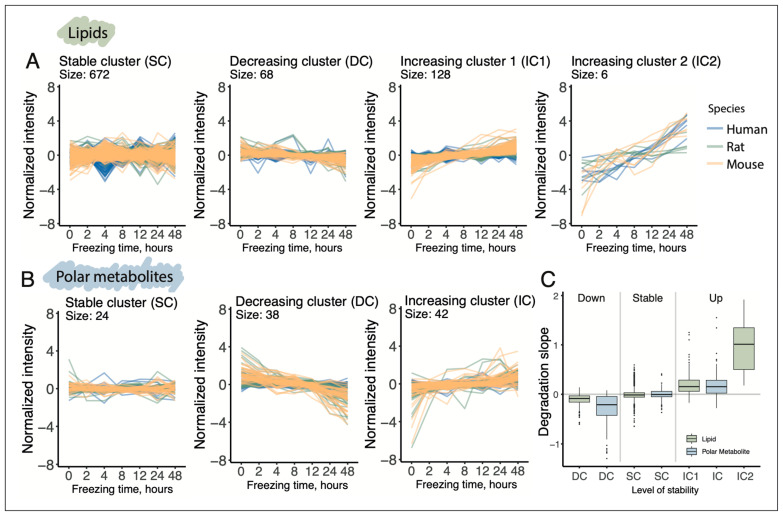
Distinct patterns of lipids and polar metabolite postmortem changes. (**A**,**B**). Clusters of lipids (**A**) and polar metabolites (**B**) based on their postmortem abundance profiles. One line represents one compound measured in samples of one species. Colors denote species, as indicated by the legend on the right. (**C**). Distributions of lipid and polar metabolite postmortem alteration rates in each of the clusters. For the individual species’ rates please see Appendix A.

**Figure 4 biomolecules-15-01288-f004:**
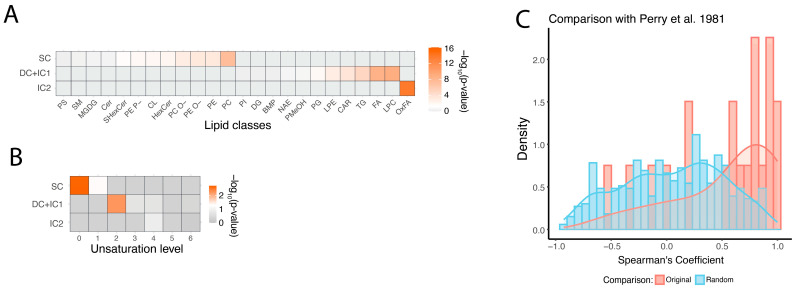
Biochemical properties of stable and unstable lipids. (**A**). Enrichment of lipids comprising stable cluster (SC), moderately unstable clusters (DC and IC1) and rapidly increasing cluster (IC2) among biochemical classes. Here and in panel B, colors show -log_10_-transformed *p*-values of the hypergeometric test, as shown by the scale bar on the right. (**B**). Enrichment of lipids comprising stable cluster (SC), moderately unstable clusters (DC and IC1) and rapidly increasing cluster (IC2) in fatty acid residues with different unsaturation extent. (**C**). Comparison of observed and published [31] postmortem metabolite level alterations. The red density plot shows the distribution of Spearman correlation coefficients for matching metabolites between the literature and the current study, while the blue plot represents the distribution generated from 10,000 permutations where the matched metabolite labels were randomly shuffled.

## Data Availability

The original contributions presented in this study are included in the article. Further inquiries can be directed to the corresponding author.

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
