# Peer review of "Postmortem Stability Analysis of Lipids and Polar Metabolites in Human, Rat, and Mouse Brains"

_biomolecules, 2025, doi:10.3390/biom15091288_

Round 1
Reviewer 1 Report
Comments and Suggestions for Authors
Postmortem stability analysis of lipids and polar metabolites in 3 human, rat, and mouse brains
Zavolskova M et al.,
This manuscript presents a very interesting and timely study that aims to fill a significant gap in the literature regarding the stability of lipids and metabolites in ex vivo tissues. The study is clearly written, well-structured, and provides valuable additional data that will be highly informative for experts interested in the temporal kinetics of molecular degradation or accumulation in rat, mouse, and human tissue samples.
The inclusion of detailed kinetic profiles for numerous species at different time points and tissue types greatly enhances the utility of the data. It provides a practical framework for researchers planning metabolomic or lipidomic studies in similar settings, especially when sample handling or storage conditions could influence molecular readouts.
The only point of clarification or potential concern concerns interindividual variability in human samples. Given the known variability in enzyme expression, metabolic activity, and genetic background among individuals, it would be useful to understand whether this variability could significantly impact the observed patterns of lipid and metabolite degradation or accumulation.
Specifically: Do the authors find notable differences among individuals in the baseline (time 0) levels of lipids and metabolites? Could this variability affect how kinetic trends are interpreted over time? In the authors' opinion, is a sample size of six individuals adequate for drawing conclusions about generalizable trends in molecular stability, or should interindividual variability be regarded as a limitation of the current study? Clarifying these issues would enhance the translational significance of the findings and aid readers in better understanding the observed patterns in human tissues.
Author Response
|
Response to Reviewer 1 Comments
|
||
|
1. Summary |
|
|
|
We are grateful to the reviewer for their thorough evaluation and comments on our manuscript.
|
||
|
3. Point-by-point response to Comments and Suggestions for Authors |
||
|
Comments 1: Do the authors find notable differences among individuals in the baseline (time 0) levels of lipids and metabolites?
|
||
|
Response 1: We appreciate the reviewer’s valuable observation regarding baseline variability. In our original analysis we did not assess the baseline variation, but we performed normalization of lipid and polar metabolite intensities for each of the timepoints to the mean intensity value of this compound in this individual to minimize base level variation among individuals within species. Now, as suggested by the reviewer, we analyzed the unnormalized intensities at the zero timepoint across the three species. As the reviewer anticipated, human samples exhibited greater variability compared to rodents, though for most compounds, the overall variability largely overlapped among species and was relatively modest in all species (see Figure R1.1 below).
Figure R1.1. Boxplots showing distributions of the standard deviation values of lipids (left) and polar metabolites (right) at zero time point. Based on the baseline variation analysis shown above and, considering the donor-based normalization procedure used in our original data processing pipeline, we suggest that individual variation effects could have substantial influence on our study results. Further, as we show in our analysis, not all compounds show reduced postmortem alteration dynamics compared to the rodents. Specifically, one lipid class – oxidized fatty acids – displayed highly consistent fast accumulation trend across all three species (Figure S3 in the manuscript also shown as Figure R1.2 below), arguing against the possibility that systematic effects potentially influencing all compounds, such as large inter-individual variation in humans, play the key role in postmortem dynamics deceleration in the postmortem human brain.
Figure R1.2. Distributions of lipid and polar metabolite postmortem alteration rates in each species for each of the clusters.
|
||
|
Comments 2: Could this variability affect how kinetic trends are interpreted over time? |
||
|
Response 2: To further assess the potential influence of inter-individual variation, we conducted a parallel analysis without donor-based normalization step. This analysis did not reveal any new unstable lipids or polar metabolites but, as it might be expected, due to retained inter-individual variation, some compounds previously classified as showing significant postmortem alterations in the original analysis no longer reached statistical significance. The effect was observed for a total of 256 compounds in total, which constitutes 90% of the original number of unstable compounds identified in the original analysis.
|
||
|
Comments 3: In the authors' opinion, is a sample size of six individuals adequate for drawing conclusions about generalizable trends in molecular stability, or should interindividual variability be regarded as a limitation of the current study? |
||
|
Response 3: We appreciate the reviewer’s comment regarding sample size and interindividual variability. We believe that including six donors per group is adequate for capturing the key trends in molecular stability in this study, as the applied donor-based normalization procedure minimized the influence of inter-individual variability. Supporting this point is the consistency of postmortem alterations observed in our study across all three species, including humans. Further, as can be seem from Figure R1.2 above (Figure S3 in the manuscript), the slope of degradation and accumulation curves for compounds that pass the significance cutoff after multiple testing correction with current sample size is already quite modest. Therefore, increasing the number of donors beyond six would likely result in statistical significance for a range of compounds displaying an even smaller amplitude of postmortem alterations. While in some cases such small changes could also be important, for most studies they might be far below biological effects in focus. In any case, information about postmortem alteration of all 978 investigated compounds, including the ones not passing the statistical significance cutoff, are included in the manuscript (Tables S4 and S5), allowing detailed inspection of the postmortem stability trends for compounds of interest. Clarifying these issues would enhance the translational significance of the findings and aid readers in better understanding the observed patterns in human tissues. We would like again to thank the reviewer for his or her comments allowing us to improve our study. We have included the above-mentioned analysis and discussion in the revised manuscript at lines 593-602.
|
||
Reviewer 2 Report
Comments and Suggestions for Authors
Marina Zavolskova and colleagues examined the effects of postmortem delay on the abundance levels of polar and hydrophobic lipid metabolites (lipids) in brain necrotic tissue samples from human (n = 6), rats (n = 6), and mice (n = 6). All the brain samples were from the temporal lobe of the cerebral cortex. Authors provided a comprehensive characterization of the postmortem instability of brain lipids and polar metabolites by analyzing their abundance levels using high performance liquid chromatography coupled with tandem mass spectrometry. The study revealed notable differences in the abundance levels of certain lipids and polar metabolites at different postmortem interval duration (up to 48h) across the three species. This is an interesting study finding providing a foundation for developing new effective strategies for some of the effects related to postmortem alterations in brain studies.
Overall, the manuscript is well written and data well-constructed and discussed.
Below are some questions that need to be addressed by the authors:
1/ What is the rational for selecting Bain rodents (rats & mice) to be compared to human samples?
2/ For the power analysis, is the size (N=6) sufficient to draw robust significance.
3/ As mentioned in the study limitations, authors used human biopsy samples collected after patient surgeries might and this might not be effective approach for a such comparative study. How do the authors control for the quality of collected tissue surrounding brain tumors in these patients. Any intra-variability or tissue heterogeneity within this group.
4/ Frontal lobe neocortex samples were dissected at the defined time point of 2, 4, 8, 12, 24, and 48 hours after dissection, placed in dry ice, then stored at -80°C until lipids/metabolites extraction. Did the authors attempt to compare tissue analysis before and after freezing them at -80°C. How were lipid samples protected from possible chemical alterations such as oxidation etc.?
Minors:
Please omit subtitle “Lipids degradation” at the beginning of discussion section
Author Response
|
Response to Reviewer 2 Comments
|
||
|
1. Summary |
|
|
|
We are grateful to the reviewer for their thorough evaluation and comments on our manuscript.
|
||
|
3. Point-by-point response to Comments and Suggestions for Authors |
||
|
Comments 1: What is the rational for selecting Bain rodents (rats & mice) to be compared to human samples? |
||
|
Response 1: We thank the reviewer for raising this point. One reason for selecting mice and rats alongside human samples was based on the simple fact that these species are the most common object in mammalian brain research (doi: 10.5555/article.2489116). Admittedly, for both mice and rats the postmortem delay does not constitute a big problem, given the controlled nature of sample collection. Further, the time scale of postmortem delay, up to 48 hours after death, does not match usual experimental setup of the rodent experiments. Thus, rodent samples were mainly used in our study to compare the postmortem alterations observed in this much more controlled model, also previously shown to display faster postmortem degradation dynamics for mRNA and proteins, with the postmortem changes observed in human samples. The results proved the applicability of this approach: both mice and rats showed faster postmortem dynamics than humans, but the direction of the changes overwhelmingly correlated among the three species. Thus, we believe, mouse and rat measurements allowed us to infer the postmortem stability properties of lipids and polar metabolites in the human brain with much greater confidence than based on the human samples alone. We have now included discussion of this point in the revised manuscript (line 610-617).
|
||
|
Comments 2. For the power analysis, is the size (N=6) sufficient to draw robust significance |
||
|
Response 2: We appreciate the reviewer’s comment regarding sample size. We believe that including six donors per group is adequate for capturing the key trends in molecular stability in this study, as the applied donor-based normalization procedure substantially minimized the influence of inter-individual variability. Supporting this point is the consistency of postmortem alterations observed in our study across all three species, including humans. Further, as can be seem from Figure R2.1 below (Figure S3 in the manuscript), the slope of degradation and accumulation curves for compounds that pass the significance cutoff after multiple testing correction with current sample size is already quite modest. Therefore, increasing the number of donors beyond six would likely result in statistical significance for a range of compounds displaying an even smaller amplitude of postmortem alterations. While in some cases such small changes could also be important, for most studies they might be far below biological effects in focus. In any case, information about postmortem alteration of all 978 investigated compounds, including the ones not passing the statistical significance cutoff, are included in the manuscript (Tables S4 and S5), allowing detailed inspection of the postmortem stability trends for compounds of interest. We certainly agree with the reviewer, however, that the exact number of compounds reported as affected by postmortem delay, does depend substantially both on the sample size and on the statistical procedures used in the study. We have added discussion of this point to the revised manuscript (line 593-602).
Figure R2.1. Distributions of lipid and polar metabolite postmortem alteration rates in each species for each of the clusters.
|
||
|
Comments 3: As mentioned in the study limitations, authors used human biopsy samples collected after patient surgeries might and this might not be effective approach for a such comparative study. How do the authors control for the quality of collected tissue surrounding brain tumors in these patients. Any intra-variability or tissue heterogeneity within this group. |
||
Response 3: We appreciate the reviewer’s valuable observation regarding potential variability in tissue quality. To ensure the reliability of the collected samples, all human biopsy specimens were carefully obtained and evaluated by experienced neurosurgeons.
Regarding baseline variability, in our original analysis we did not assess it, but we performed normalization of lipid and polar metabolite intensities for each of the timepoints to the mean intensity value of this compound in this individual to minimize base level variation among individuals within species. Now, we analyzed the unnormalized intensities at the zero timepoint across the three species. As it might be expected, human samples exhibited somewhat greater variability compared to rodents, though for most compounds, the overall variability largely overlapped among species and was relatively modest in all species (see Figure R2.2 below). Based on this analysis and, considering the donor-based normalization procedure used in our original data processing pipeline, we suggest that individual variation effects could have substantial influence on our study results. Further, as we show in our analysis, not all compounds show reduced postmortem alteration dynamics compared to the rodents. Specifically, one lipid class – oxidized fatty acids – displayed highly consistent fast accumulation trend across all three species (Figure S3 in the manuscript also shown as Figure R2.1 above), arguing against the possibility that systematic effects potentially influencing all compounds, such as large inter-individual variation in humans, play the key role in postmortem dynamics deceleration in the postmortem human brain. We have clarified discussion of this point in the revised manuscript (line 603-609).

Figure R2.2. Boxplots showing distributions of the standard deviation values of lipids (left) and polar metabolites (right) at zero time point.
Comments 4: Frontal lobe neocortex samples were dissected at the defined time point of 2, 4, 8, 12, 24, and 48 hours after dissection, placed in dry ice, then stored at -80°C until lipids/metabolites extraction. Did the authors attempt to compare tissue analysis before and after freezing them at -80°C. How were lipid samples protected from possible chemical alterations such as oxidation etc.?
Response 4: We thank the reviewer for raising this important point. In this study, we did not perform direct comparisons of tissue analysis before and after freezing at −80°C. However, snap-freezing and storage at −80°C are well-established methods to minimize post-collection biochemical alterations, including oxidation. While we did not experimentally validate this for brain tissue in our work, similar stability assessments in other biological substances (e.g., plasma) support the robustness of this approach. For instance:
Angela M. Zivkovic et al. (2009) (Metabolomics) demonstrated that multiple freeze-thaw cycles of plasma did not significantly alter metabolite profiles (DOI: 10.1007/s11306-009-0174-2).
Mark Haid et al. (2017) (J. Proteome Res.) reported high stability of plasma metabolites after long-term storage at −80°C (DOI: 10.1021/acs.jproteome.7b00518).
In our study, all samples were stored at −80°C for less than one month before processing, further reducing risks of degradation. To address potential oxidation, tissues were flash-frozen immediately after dissection, and lipids were extracted under controlled conditions. That said, we fully agree that systematic evaluation of freezing effects on brain metabolomics remains underexplored and warrants future investigation. We have added a brief discussion of this point in the revised manuscript (line 644-651) to highlight both the study limitation and the need for further methodological standardization in the field.
Minors:
Please omit subtitle “Lipids degradation” at the beginning of discussion section
We thank the reviewer for pointing out this issue; it has now been fixed.
We would like again to thank the reviewer for his or her comments allowing us to improve our study.
Reviewer 3 Report
Comments and Suggestions for Authors
The manuscript examined the postmortem stability of lipids and polar metabolites over a 48-hour interval in the brains of three species—humans, rats, and mice—and found that the abundance levels of 23% of the 867 studied lipids and 75% of the 104 studied polar metabolites were significantly affected postmortem in at least one species. The postmortem effects correlated positively and significantly among species, with an approximately tenfold slower rate in humans compared to rodents. However, the study has several issues that should be addressed.
A. In most biochemical, metabolomic, and molecular biology experiments, freshly collected tissues are immediately snap-frozen in liquid nitrogen and subsequently stored at −80°C to minimize enzymatic degradation and metabolic alterations. In this study, samples were stored at +4°C until frozen on dry ice after dissection and then examine the postmortem stability of lipids and polar metabolites. The authors should clarify the rationale for using 4°C storage, and discuss why they did not investigate whether storage at −80 °C preserves lipid and metabolite levels in a manner that accurately reflects the state at the time of death. Then, if the aim is solely to examine postmortem stability of these metabolites, why not also consider other temperatures, such as 20 °C?
B. Compound identification was based entirely on an internal database and manual verification. To strengthen confidence in the method’s reliability, the authors should validate several representative compounds using authentic external standards.
C. The statistical analysis employs linear modeling of metabolite intensities over logâ‚‚-transformed freezing times with Bonferroni correction, followed by k-means clustering of unstable metabolites, which is conceptually sound; however, the assumption of linearity may not capture potentially non-linear postmortem changes. The small sample size (n = 6 per time point), combined with the conservative Bonferroni adjustment, may reduce statistical power and increase the risk of type II errors. Donor-mean normalization could mask absolute concentration changes relevant to PMI interpretation, and the choice of cluster number based solely on the elbow method without biological priors may limit the robustness of pattern classification. Alternative approaches, such as non-linear fitting, less conservative multiple-testing corrections, or mixed-effects models, could strengthen the analysis.
Minor comments:
Line 117: “1,5 ml microtubes” should be corrected to “1.5 ml microtubes.”
Lines 325–326: “at two, four, eight, 12, 24, and 48 hours post-surgery or post-sacrifice (Fig. 1A)”—the style of numerical descriptions should be consistent.
The manuscript states that UPLC-MS spectral data were saved in .raw format, converted to .abf format using the ABF converter, and processed with MS-DIAL (v4.90). The storage format information is unnecessary and can be omitted.
Author Response
|
Response to Reviewer 3 Comments
|
||
|
1. Summary |
|
|
|
We are grateful to the reviewer for their thorough evaluation and comments on our manuscript.
|
||
|
3. Point-by-point response to Comments and Suggestions for Authors
|
||
|
Comments 1: In most biochemical, metabolomic, and molecular biology experiments, freshly collected tissues are immediately snap-frozen in liquid nitrogen and subsequently stored at −80°C to minimize enzymatic degradation and metabolic alterations. In this study, samples were stored at +4°C until frozen on dry ice after dissection and then examine the postmortem stability of lipids and polar metabolites. The authors should clarify the rationale for using 4°C storage, and discuss why they did not investigate whether storage at −80 °C preserves lipid and metabolite levels in a manner that accurately reflects the state at the time of death. Then, if the aim is solely to examine postmortem stability of these metabolites, why not also consider other temperatures, such as 20 °C? Response 1: We thank the reviewer for brining attention to this point. When studying postmortem brain tissue, we must consider that, prior to dissection, the intact human brain typically remains at the original body temperature and then at room temperature for several hours after death. To assess postmortem changes at different time points, however, we must disrupt this intact state of the brain tissue during sample collection, which may itself alter degradation rates. In this study, we selected +4°C storage as a compromise: it allows us to approximate real-world conditions (where tissue is exposed to non-freezing temperatures) while mitigating the extreme degradation that would occur at room temperature in the brain samples with integrity compromised by dissection. Admittedly, this approach cannot fully replicate the intact postmortem environment, and we have now added this limitation or our study to revised manuscript (line 636-644). Nevertheless, the approach we took provides a controlled framework to evaluate the directional trends and relative stability of both lipids and poler metabolites in the postmortem brain.
|
||
|
Comments 2. Compound identification was based entirely on an internal database and manual verification. To strengthen confidence in the method’s reliability, the authors should validate several representative compounds using authentic external standards. |
||
|
Response 2: We thank the reviewer for highlighting this place. We should describe the annotation process more consistently. At the first stage, we used the MSDial databases, then we checked them against the internal database, which also contains information about the internal labeled standards that were added. At the last step, we manually checked the correctness of the integration, since especially for isomers with close retention times, the integration may be inaccurate. We have rewritten this part in the text. Below you can see the spectra of some compounds and their labeled analogues. You can see that the retention times and fragmentation spectra correspond to each other. Methionine: Retention time
Methionine: fragmentation
Phosphatidylethanolamine: Retention time |
||
![]() |
||
|
Phosphatidylethanolamine: fragmentation |
||
![]() |
||
|
Figure R3.2. Annotation validation using methionine and phosphatidylethanolamine as examples
|
||
|
Comments 3: The statistical analysis employs linear modeling of metabolite intensities over logâ‚‚-transformed freezing times with Bonferroni correction, followed by k-means clustering of unstable metabolites, which is conceptually sound; however, the assumption of linearity may not capture potentially non-linear postmortem changes. The small sample size (n = 6 per time point), combined with the conservative Bonferroni adjustment, may reduce statistical power and increase the risk of type II errors. Donor-mean normalization could mask absolute concentration changes relevant to PMI interpretation, and the choice of cluster number based solely on the elbow method without biological priors may limit the robustness of pattern classification. Alternative approaches, such as non-linear fitting, less conservative multiple-testing corrections, or mixed-effects models, could strengthen the analysis. |
||
Response 3:
We thank the reviewer for bringing attention to this issue. We acknowledge that the potential for type II errors is indeed a valid concern given the relatively small sample size and conservative Bonferroni correction. To address this point, we repeated our analysis using the less conservative Benjamini-Hochberg correction. As it might be expected, we identified additional compounds passing this significance threshold. The amplitudes of their postmortem changes were, on average, 10-fold smaller than those selected under the Bonferroni adjustment. To illustrate this result, we show as an example the distributions of the absolute change values normalized by the compound intensities, for mouse brain lipids (Figure R3.2 below). Thus, use of the Bonferroni correction in our statistical analysis allows to single out compounds not only showing the statistical significance but also displaying substantial amplitude of postmortem alterations. At the same time, we understand that in some cases small amplitude postmortem alterations could also be important. Therefore, we included information about postmortem alteration of all 978 investigated compounds, including the ones not passing the statistical significance cutoff, in the manuscript (Tables S4 and S5), thus allowing detailed inspection of the postmortem stability trends for compounds of interest. We have now included a more detailed discussion of this point to the revised manuscript (line 586-592).

Figure R3.2. The distributions of the absolute change values normalized by the compound intensities of mouse brain lipids passing the Bonferroni correction (red) and the additional ones passing the Behjamini-Hochberg correction (blue).
Further, as suggested by the reviewer, we evaluated the assumption of linearity of postmortem alterations with log-transformed postmortem interval duration time by testing Spearman’s rank correlation, insensitive to the linearity assumption, as an alternative to Pearson’s linear correlation. As can be seem from the tables below, we observed only minimal changes in the numbers of detected unstable lipids and polar metabolites, except the loss of statistical power in metabolite analysis when Spearman’s rank correlation test was coupled with strict Bonferroni correction.
|
Mouse; Lipids |
Pearson |
Spearman |
|
Before adjustment |
329 |
308 |
|
Benjamini-Hochberg |
304 |
306 |
|
Bonferroni |
166 |
165 |
|
Rat; Lipids |
Pearson |
Spearman |
|
Before adjustment |
346 |
359 |
|
Benjamini-Hochberg |
314 |
319 |
|
Bonferroni |
135 |
137 |
|
Human; Lipids |
Pearson |
Spearman |
|
Before adjustment |
93 |
98 |
|
Benjamini-Hochberg |
5 |
5 |
|
Bonferroni |
5 |
5 |
|
Mouse; Metabolites |
Pearson |
Spearman |
|
Before adjustment |
81 |
80 |
|
Benjamini-Hochberg |
79 |
74 |
|
Bonferroni |
58 |
0 |
|
Rat; Metabolites |
Pearson |
Spearman |
|
Before adjustment |
74 |
77 |
|
Benjamini-Hochberg |
72 |
75 |
|
Bonferroni |
58 |
0 |
|
Human; Metabolites |
Pearson |
Spearman |
|
Before adjustment |
20 |
20 |
|
Benjamini-Hochberg |
10 |
6 |
|
Bonferroni |
5 |
0 |
To assess the robustness of our clustering procedure, we applied unsupervised clustering approach – hierarchical clustering implemented in R heatmap algorithm. The results confirmed division to two main clusters, ascending and descending ones, for polar metabolites and the presence of an additional cluster, a rapidly ascending one, for lipids (see Figure R3.3 below). As a side note, we found a typo in the methods description, where it is written that for lipids the optimal number of clusters is 4, corrected to 3 now.

Figure R3.3. Unsupervised clustering of lipids (left) and polar metabolites (right)
Minors:
Line 117: “1,5 ml microtubes” should be corrected to “1.5 ml microtubes.”
We thank the reviewer for pointing out this error; it has now been corrected.
Lines 325–326: “at two, four, eight, 12, 24, and 48 hours post-surgery or post-sacrifice (Fig. 1A)”—the style of numerical descriptions should be consistent.
We thank the reviewer for pointing out this typo; it has now been corrected.
The manuscript states that UPLC-MS spectral data were saved in .raw format, converted to .abf format using the ABF converter, and processed with MS-DIAL (v4.90). The storage format information is unnecessary and can be omitted.
We thank the reviewer for pointing out this detail; it has now been corrected.
We would like again to thank the reviewer for his or her comments allowing us to improve our study.
Round 2
Reviewer 2 Report
Comments and Suggestions for Authors
Authors have satisfactorily addressed all the questions raised in the first round review.
Reviewer 3 Report
Comments and Suggestions for Authors
I appreciate the revisions you have made to the manuscript.